# Usefulness of Longitudinal Strain Adjusted to Regional Thickness in Hypertrophic Cardiomyopathy

**DOI:** 10.3390/jcm11082089

**Published:** 2022-04-08

**Authors:** Sophie Urtado, Hélène Hergault, Stephen Binsse, Vincent Aidan, Mounir Ouadahi, Catherine Szymanski, Sophie Mallet, Marie Hauguel-Moreau, Robert Yves Carlier, Olivier Dubourg, Nicolas Mansencal

**Affiliations:** 1Department of Cardiology, Ambroise Paré Hospital, Assistance Publique-Hôpitaux de Paris (AP-HP), Centre de Référence des Cardiomyopathies et des Troubles du Rythme Cardiaque Héréditaires ou Rares, Université de Versailles-Saint Quentin (UVSQ), 92100 Boulogne-Billancourt, France; urtado.sophie@outlook.fr (S.U.); helene.hergault@aphp.fr (H.H.); vincent.aidan@aphp.fr (V.A.); mounir.ouadahi@aphp.fr (M.O.); catherine.szymanski@aphp.fr (C.S.); sophie.mallet@aphp.fr (S.M.); marie.hauguel@aphp.fr (M.H.-M.); olivier.dubourg@aphp.fr (O.D.); 2INSERM U-1018, Centre de Recherche en Epidémiologie et Santé des Populations (CESP), Clinical Epidemiology Team, UVSQ, 94800 Villejuif, France; 3Department of Radiology, Ambroise Paré Hospital, AP-HP, 92100 Boulogne-Billancourt, France; stephen.binsse@aphp.fr (S.B.); robert.carlier@aphp.fr (R.Y.C.); 4INSERM U-1179, Handicap Neuromusculaire, UVSQ Paris-Saclay, 92380 Garches, France

**Keywords:** hypertrophic cardiomyopathy, strain, fibrosis

## Abstract

Background. We assessed the usefulness of a longitudinal strain adjusted to regional thickness in hypertrophic cardiomyopathy (HCM). Indeed, with conventional software, the width of the region of interest (ROI) is the same over the entire myocardial wall, wherein the software analyzes only partially the left ventricular (LV) hypertrophic segments. Methods. We included 110 patients: 55 patients with HCM (HCM group) and 55 healthy subjects (age- and sex-matched control group). The global longitudinal strain (GLS) and regional strain for each of the 17 segments was calculated with standard software (for two groups) and with software adjusted to the myocardial wall thickness (for the HCM group). Results. GLS was significantly decreased in the HCM group compared to the control group (−15.1 ± 4.8% versus −20.5 ± 4.3%, *p* < 0.0001). In the HCM group, GLS (standard method versus adjusted to thickness) measurements were not significantly different (*p* = 0.34). Interestingly, the regional strain adjusted to thickness was significantly lower than the standard strain in the hypertrophic segments, especially in the basal inferoseptal segment (*p* = 0.0002), median inferoseptal segment (*p* < 0.001) and median anteroseptal segment (*p* = 0.02). The strain adjusted to thickness was still significantly lower in the most hypertrophic segments (≥20 mm) (−3.7 ± 3%, versus −5.9 ± 4.4%, *p* = 0.049 in the basal inferoseptal segment and −5.7 ± 3.5% versus −8.3 ± 4.5%, *p* = 0.0007 in the median inferoseptal segment). In the segments with significant myocardial fibrosis, the longitudinal strain adjusted to thickness was significantly lower than the conventional strain (−8.3 ± 3.3% versus −11.4 ± 4.5%, *p* = 0.002). The analysis of the strain adjusted to thickness had a better feasibility (97.5% versus 99%, *p* = 0.01). Conclusions. The analysis of a longitudinal strain adjusted to regional thickness is feasible in HCM and allows a better evaluation of myocardial deformation, especially in the most LV hypertrophic segments.

## 1. Introduction

Cardiomyopathies are usually divided into four main subtypes [1]: hypertrophic cardiomyopathy, dilated cardiomyopathy, restrictive cardiomyopathy and arrhythmogenic right ventricular cardiomyopathy, with other non-classified subtypes. Hypertrophic cardiomyopathy (HCM) is a heart disease defined by the presence of increased left ventricular (LV) wall thickness without abnormal loading conditions (hypertension, aortic stenosis) [1,2,3,4]. Its prevalence ranges from 0.02% to 0.23% [1]. In around 50% of HCM, the disease is an autosomal dominant trait (mutations in cardiac sarcomere protein genes). In 5–10% of cases, other genetic or non-genetic causes may induce HCM, such as glycogen storage diseases, Anderson–Fabry disease, Friedrich’s ataxia, malformation syndromes, amyloidosis, mitochondrial diseases, etc. [1]. Histologically, HCM is characterized by myocardial disarray, the hypertrophy of heart muscle fibers and interstitial fibrosis. These structural abnormalities lead to the early impairment of diastolic function, even without myocardial hypertrophy or symptoms [5]. However, the alteration of systolic function is a real challenge to evaluate, because the LV ejection fraction usually seems to be preserved [6,7]. A better understanding of this complex physiopathology allows us to optimize the management of patients and also of their relatives [8].

New imaging techniques, such as myocardial deformation, have been developed for the assessment of LV function. In HCM, longitudinal strain analysis allows us to detect early LV contraction abnormalities despite obvious preserved LV ejection fractions [4,9], and some studies suggest a prognostic interest in cardiovascular risk stratification in these patients [10,11].

With previous software, the width of the region of interest (ROI) is the same over the entire myocardial wall. Unfortunately, in HCM, hypertrophy is usually asymmetrical and can be localized to a few segments, leading to a partial assessment of LV thickness in HCM. The aim of this study is to evaluate a software for strain analysis with an adjustable ROI according to each segment thickness, in comparison with standard software.

## 2. Materials and Methods

We studied 110 patients divided into 2 groups: 55 patients admitted in our reference center for inherited heart disease with the confirmed diagnosis of HCM (HCM group) and 55 healthy subjects (age- and sex-matched control group).

In the HCM group, entry criteria included a definitive diagnosis of HCM, established by 3 experts, according to personal and familial history, physical examination, echocardiography, cardiac magnetic resonance (CMR) and genetic data. Epidemiological, clinical status and echocardiographic data were collected for each patient. Sudden cardiac death 5-year risk score [1,12] was calculated for each patient and we assessed five major risk factors (personal history of syncope, family history of premature HCM-related sudden death, maximal LV wall thickness ≥ 30 mm, abnormal blood pressure response to exercise and non-sustained ventricular tachycardia) [3]. In the control group, all subjects had neither risk factors nor evidence of cardiovascular disease.

All patients and subjects included in the two groups (HCM and control groups) underwent transthoracic echocardiographic examination and the global longitudinal strain was systematically calculated for each patient. All echocardiographic examinations were performed using a Vivid 9 system (GE Medical Systems, Horten, Norway) and were digitally recorded. All measurements were performed with EchoPAC software (GE Medical Systems, Horten, Norway) according to the recommendations [1,13] and were averaged over 3 cardiac cycles. Several two-dimensional views were routinely recorded: parasternal long-axis and short-axis views and apical two-, three- and four-chamber views. The following two-dimensional measurements were systematically assessed: (1) end-diastolic measurements of the interventricular septum, posterior wall and left ventricle; (2) end-systolic LV diameter; (3) left atrial diameter and volume; (4) Maron index. We systematically assessed mitral regurgitation (and its severity) and systolic anterior motion (SAM) of the anterior mitral leaflet. Left ventricular outflow tract obstruction was scanned with a continuous wave Doppler to measure maximal outflow velocity, at rest and after a Valsalva maneuver. The phenotype of hypertrophic cardiomyopathy was defined according to the Maron classification: type I as hypertrophy involving the basal septum, type II as hypertrophy of the whole septum, type III as hypertrophy involving the septum, anterior, and anterolateral walls and type IV as apical hypertrophy.

We analyzed myocardial deformation of the LV from two-dimensional apical four-, two- and three-chamber views by using speckle tracking. Two-dimensional images of a cardiac cycle were registered with a low-frequency acoustic signal (70–80 Hz). The left ventricle was divided into 17 segments and each segment was analyzed separately. Global longitudinal strain (GLS) and regional strain for each of the 17 segments was calculated with standard software (for the HCM and control groups) and with software adjusted to myocardial wall thickness (for the HCM group).

The measurement of global and regional longitudinal standard strain was performed using automated function imaging (AFI) method from apical two-, three- and four-chamber views by an independent operator. Myocardial contouring was verified and corrected manually by adjusting the ROI. The ROI was identical for each segment analyzed (Figure 1A). A global bull’s eye representation was generated with data obtained for each segment, and the average value of peak systolic strains constituted the global longitudinal strain.

The longitudinal strain adjusted to myocardial wall thickness was measured using the same apical two-, three- and four-chamber views. An independent operator traced the endocardial contour, and the closure of aortic valve was signaled to the software, defining end-systole. The ROI was manually enlarged to fit the internal and external border with the myocardial wall, allowing an analysis of the entire hypertrophic myocardium even if there was asymmetric thickness between the different segments (Figure 1B). For each patient, a strain analysis of 17 LV segments was obtained. Values of myocardial deformation were represented as a curve, and the peak of end-systolic strain was recorded for each segment. The average of these values constituted global longitudinal strain. Moreover, the myocardial wall was divided into 3 layers: the endocardial, mid-wall and epicardial layers. Longitudinal strain was measured for each segment of these 3 layers.

CMR was performed in all patients presenting with HCM with a 1.5-T MRI system (Optima MR450w, GE Healthcare, Milwaukee, WI, USA). CMR images acquired on the long axis and short axis served to assess global and regional LV function. The cardiac short-axis sequences were planned to cover the entire left ventricle using contiguous 8 mm-thick slices. A bolus of 0.1 mM gadolinium-based contrast agent (Dotarem^®^, Guerbet, France, 0.1 mmol/kg) was injected at a rate of 4.0 mL/s. Ten minutes after contrast injection, breath-hold contrast-enhanced 3D T1-weighted inversion recovery gradient echo sequences were acquired to detect late gadolinium enhancement (LGE). Significant myocardial fibrosis was systematically assessed and was defined as an LGE higher than 15% of left ventricular mass [14]. The study was approved by the Institutional Data Protection Authority of Paris Saclay University Hospitals.

Statistical analysis was performed using StatView software version 4.5 (Abacus Concepts, Inc., Berkeley, CA, USA). Continuous variables are presented as mean ± SD (standard deviation) and categorical data are presented as absolute values and percentages. Categorical variables were compared with the use of the Mc Nemar test. Continuous variables were compared with the use of a paired t test. Linear regression was used to compare strain values, echocardiographic data and sudden cardiac risk score. Concordances between standard strain and strain adjusted to the myocardial wall thickness were evaluated using Bland–Altman plot. A value of *p* < 0.05 was considered to be statistically significant.

## 3. Results

### 3.1. Clinical Characteristics

Fifty-five HCM patients (mean age 47 ± 19 years), and fifty-five healthy subjects (mean age 47 ± 19 years) were included. The clinical characteristics of HCM group and control group are presented in Table 1. In the HCM group, six patients (11%) were in New York Heart Association (NYHA) functional class III–IV versus all subjects from the control group, who were in NYHA functional class I (*p* < 0.0001). In the HCM group, a genetic mutation was found in twenty-three patients (42%) and six patients had an implantable cardioverter defibrillator or a pacemaker.

### 3.2. Echocardiographic Parameters

Echocardiographic parameters are presented in Table 2. Mean maximal myocardial thickness was 19.1 ± 6.4 mm. A significant ventricular gradient was present in 16 patients (29%, 42 ± 20 mmHg; range: 10–99 mmHg).

### 3.3. Myocardial Deformation

Mean conventional GLS was −15.5 ± 4.2% in HCM patients and −20.5 ± 4.3% in control group (*p* < 0.0001). Using the measurement of strain adjusted for myocardial wall thickness, we observed a significant gradient of myocardial deformation from the endocardial layer to the epicardial layer (−17.3 ± 5.1% for endocardial layer, −15.1 ± 4.8% for mid-wall layer and −12.8 ± 4.6% for epicardial layer, *p* < 0.0001). A significant difference between standard GLS and adjusted GLS in epicardial and endocardial layer was observed (*p* < 0.001 for both). However, GLS for the standard method versus mid-wall layer were not significantly different (*p* = 0.34). GLS was lower in type 3 pattern and type 4 pattern than in the type 1 pattern (*p* = 0.009 and *p* = 0.04, respectively). In each subgroup according to the Maron classification, there was no significant difference for GLS measured with standard method or with the strain adjusted to the myocardial wall thickness.

### 3.4. Assessment of Regional Longitudinal Strain

Results are presented in Table 3. Regional strain adjusted for thickness (mid-wall layer) was significantly lower than the conventional strain in hypertrophic segments (Figure 2): in the basal inferoseptal segment (*p* = 0.0002), in the median inferoseptal segment (*p* < 0.001) and in the median anteroseptal segment (*p* = 0.02). No significant decrease in the strain adjusted for thickness was observed in the basal anteroseptal segment versus standard strain (*p* = 0.067). Significant correlations were observed between maximum wall thickness and GLS with both the conventional measurement (r = 0.42, *p* = 0.002) and when adjusted for myocardial wall thickness (r = 0.40, *p* = 0.003). However, linear correlations were stronger when studying the regional strain of hypertrophic segments, especially in the basal inferoseptal segment (r = 0.55, *p* < 0.0001 with adjusted strain versus r = 0.48 with conventional strain, *p* = 0.0002). The best correlation was obtained when combining the regional strain of both basal and median inferoseptal segments (r = 0.61, *p* < 0.0001 with adjusted strain versus r = 0.58, *p* < 0.0001 with conventional strain). In hypertrophic segments, the gradient of myocardial deformation from the endocardial layer to the epicardial layer was significantly lower (−2.5 ± 1.7% versus −6.9 ± 3.1% in segments without hypertrophy, *p* < 0.001).

### 3.5. Analysis in Case of Hypertrophy > 20 mm

Fifteen patients had a maximal septal thickness above 20 mm. The regional strain adjusted for thickness was significantly lower in the basal inferoseptal segment (−3.7 ± 3%, versus −5.9 ± 4.4% for the standard strain, *p* = 0.049) and in the median inferoseptal segment (−5.7 ± 3.5% versus −8. ± 4.5% for the standard strain, *p* = 0.0007).

### 3.6. Global and Regional Longitudinal Strain and Cardiac Magnetic Resonance

Significant myocardial fibrosis was present in 17 patients (31%). In the segments with significant myocardial fibrosis, the longitudinal strain adjusted for thickness was significantly lower, as compared to the conventional strain (−8.3 ± 3.3% versus −11.4 ± 4.5%, *p* = 0.002).

### 3.7. Relations between Longitudinal Strain and Evaluated Risk of Sudden Death

When it was applicable (n = 49), the mean five-year sudden cardiac death risk score was 3.56 ± 7.06%: 40 patients (73%) had a score below 4%, 7 patients (13%) had a score between 4% and 6%, and 2 patients (4%) had a score above 6%. There was no correlation between strain adjusted to thickness and 5-year sudden cardiac death risk score, neither for GLS nor for the regional strain of the most hypertrophic segments. Among the five major risk factors, twenty-two patients had no risk factor, twenty patients had one risk factor and three patients had two risk factors. No difference of GLS was observed between the two strain methods. In the basal inferoseptal segment, the longitudinal strain was significantly lower in patients with at least one risk factor (−9.3 ± 3.3% versus −11.4 ± 4.5% in patients without a risk factor, *p* = 0.02).

### 3.8. Feasibility

The measurement of the standard strain was feasible in 912 segments out of 935 (97.5% of segments analyzed). The measurement of the strain adjusted to thickness was feasible in 926 segments out of 935 (99% segments analyzed, *p* = 0.01).

## 4. Discussion

In the present study, we evaluated the usefulness of a longitudinal strain adjusted to regional thickness in hypertrophic cardiomyopathy. We found that analyzing the strain adjusted for thickness is feasible and the obtained values are lower in the case of severe hypertrophy in septal segments and of significant myocardial fibrosis in patients presenting with HCM.

Myocardial deformation techniques analyze intrinsic properties of the cardiac muscle, and strain analysis has enabled advances in the understanding of HCM. Serri et al. [9] demonstrated that global and regional strains significantly decreased in patients presenting with HCM, despite visually preserved LVEF. Thus, this facilitates an early detection of the impairment of LV systolic function in HCM. Yang et al. [15] highlighted the link between the degree of hypertrophy and regional strain and found that strain was significantly lower and even reversed in the mid-septum as compared to other segments. Furthermore, longitudinal deformation was related to the degree of histopathologic abnormalities (myocyte hypertrophy, disarray, interstitial fibrosis and small intracoronary arteriole dysplasia) [16]. Despite these important findings, these studies evaluated only a part of hypertrophy, because the width of the ROI was constant and could not be adjusted to myocardial thickness, even in the case of major hypertrophy. To our knowledge, our study is the first to evaluate a technique of longitudinal strain with an adjustable region of interest. Our results are consistent with previous studies in HCM [9]. Global longitudinal strain was impaired in HCM patients, with a standard method and with the adjusted to myocardial thickness method. Moreover, when analyzing hypertrophic segments, we found that regional strain was significantly lower when adjusted for myocardial thickness. The correlation between maximum wall thickness and regional strain was even stronger when analyzing the entire hypertrophic septum. The results were particularly significant for the most severe hypertrophic segments (above 20 mm) and we also found a more severe decrease in the regional strain in myocardial segments presenting with significant fibrosis. For each pattern of HCM, the basal septum strain was significantly decreased using the two techniques of myocardial deformation, but the strain adjusted to myocardial thickness had the lowest values. Some authors studied links between strain and prognosis in HCM. Saito et al. [10] found a correlation between low global longitudinal strain and myocardial fibrosis identified using magnetic resonance imaging. Interestingly, regional peak systolic longitudinal strain was lower in segments with late gadolinium enhancement. Several authors underlined the link between lower GLS and major cardiac events, particularly ventricular arrhythmias [10,11,17,18]. However, when we assessed the sudden cardiac death risk score, no correlation was found with longitudinal strain, neither with GLS nor with the most hypertrophied segments. This result may be explained by a high proportion of patients presenting with a low sudden cardiac death risk scores and by the fact that fibrosis is not included in this score.

A three-layer analysis of longitudinal strain allows the independent quantification of myocardial deformation from the endocardial layer to the epicardial layer. In healthy subjects, there is a myocardial deformation gradient within the three layers: the longitudinal strain is higher in the endocardium as compared to the epicardium [19]. In HCM patients, Di Bella et al. [20] showed that longitudinal strain was lower in the endocardial and epicardial layers than in healthy subjects. In our study, we also found the GLS to be impaired in the three layers of the myocardial wall, with a myocardial deformation gradient between the endocardial and epicardial strains.

The bidimensional strain adjusted to myocardial thickness is a simple tool to use and it is reproducible. The mean time to obtain results is longer than with the standard strain, but this limitation is inherent to the technique, since the operator must choose, for each view, the width of the ROI. However, tracking is better with this new tool, with better feasibility.

One of the limitations of this study is the small number of patients, generating low statistical power. However, we compared our patients with HCM to a control group and, furthermore, we were able to compare CMR [21,22] and two different strain techniques, which is a recent powerful echocardiographic tool. Moreover, the use of myocardial deformation techniques requires an optimal imaging quality [23,24]. In our study, all patients had an adequate echogenicity to use the strain method, and the strain method can be less reliable in the case of poor echogenicity. Finally, we did not analyze the circumferential and radial components of deformation because their accuracy is questionable [25,26]. We preferred to focus on a practical and reproducible strain that can be used in routine practice.

## 5. Conclusions

The assessment of the complete myocardial deformation in HCM is of interest. We demonstrate that the longitudinal strain adjusted to regional thickness in hypertrophic cardiomyopathy is lower than the standard strain, mainly in the most hypertrophied segments and in the case of significant fibrosis. This evaluation could be useful for a better evaluation of prognosis, and other studies are necessary to explore these links.

## Figures and Tables

**Figure 1 jcm-11-02089-f001:**
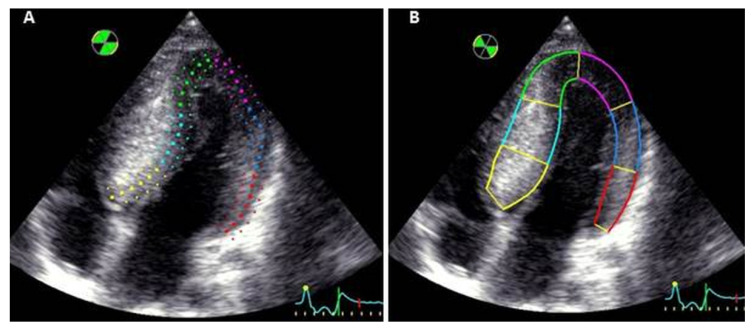
Myocardial contouring in the same apical four-chamber view (**A**): with standard software and (**B**): with software adjusted to the myocardial wall thickness.

**Figure 2 jcm-11-02089-f002:**
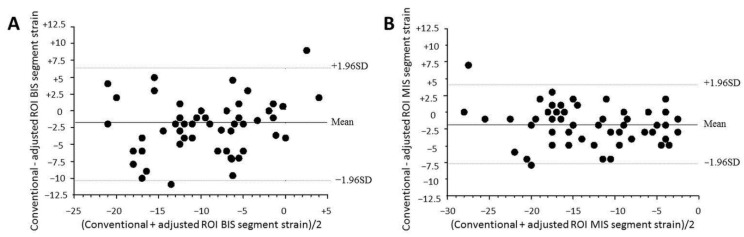
Bland–Altman plots of (**A**) basal inferoseptal (BIS) segment strain and (**B**) median inferoseptal segment strain for conventional and mid-wall adjusted ROI strains. Dashed lines, 1.96 SDs from the mean. BIS: basal inferoseptal, MIS: median inferoseptal, ROI: region of interest.

**Table 1 jcm-11-02089-t001:** Clinical characteristics of HCM group and control group.

	HCM Group n = 55	Control Group n = 55
Men, n (%)	38 (69%)	38 (69%)
Mean age, years	47 ± 19	47 ± 19
Previous history		
Syncope, n (%)	6 (11%)	0 (0%)
Chest pain, n (%)	4 (7%)	0 (0%)
Family history of sudden cardiac death, n (%)	6 (11%)	0 (0%)
Supraventricular tachycardia, n (%):	13 (24%)	0 (0%)
Atrial fibrillation	9 (16%)	0 (0%)
Atrial flutter	1 (2%)	0 (0%)
Wolff–Parkinson–White syndrome	2 (4%)	0 (0%)
Dyspnea (NYHA classification) *		
Class I, n (%)	22 (56%)	55 (100%)
Class II, n (%)	18 (32%)	0 (0%)
Class III, n (%)	6 (11%)	0 (0%)
Class IV, n (%)	0	0 (0%)
Heart rate, /min	68 ± 13	75 ± 11
Systolic blood pressure, mmHg	122 ± 20	125 ± 11
Diastolic blood pressure, mmHg	73 ± 10	74 ± 8
NSVT on Holter ECG, n (%)	5 (9%)	NA
Hypotensive blood pressure response to exercise, n (%)	2 (4%)	NA
Identified gene mutation, n (%)	23 (42%)	NA
Myosin-binding protein C (MYBPC3), n (%)	15 (27%)	
Beta-myosin heavy chain (MYH7), n (%)	6 (11%)	
Cardiac troponin T (TNNT2), n (%)	1 (2%)	
Protein kinase A, gamma unit (PKRAG2), n (%)	1 (2%)	
Significant myocardial fibrosis on CMR, n (%)	17 (31%)	NA
Medication		
Beta-blockers, n (%)	37 (67%)	0 (0%)
ACE inhibitor/ARA, n (%)	3 (5%)	0 (0%)
Diuretics, n (%)	10 (18%)	0 (0%)
Disopyramide, n (%)	6 (11%)	0 (0%)
Amiodarone, n (%)	2 (4%)	0 (0%)
Flecainide, n (%)	2 (4%)	0 (0%)
Dual-chamber pacemaker implantation/ICD, n (%)	6 (11%)	0 (0%)
Septal alcohol ablation, n (%)	5 (9%)	-
Myoto myomectomy, n (%)	2 (4%)	-
5-year sudden cardiac death risk score, n (%)	3.56 ± 7.06	NA

ACE inhibitor/ARA, angiotensin-converting enzyme inhibitor/angiotensin II receptor antagonist; HCM, hypertrophic cardiomyopathy; ICD, implantable cardioverter defibrillator; CMR, cardiac magnetic resonance; NA, not applicable; NSVT, non-sustained ventricular tachycardia; NYHA, New York Heart Association. * *p* < 0.001 between HCM group and control group.

**Table 2 jcm-11-02089-t002:** Echocardiographic characteristics of HCM and control groups.

	HCM Group n = 55	Control Group n = 55
LV diastolic diameter (mm) *	44.6 ± 5.9	48.7 ± 5.2
Maximal septal thickness (mm) *	19.1 ± 6.4	7.9 ± 1.5
LV ejection fraction	60 ± 11	62 ± 3
Septum/posterior wall ratio *	1.89 ± 0.73	1.05 ± 0.02
Maron index (mm)	56.8 ± 14.4	NA
Systolic anterior motion, n (%)	27 (49%)	0 (0%)
Maximal left ventricular gradient		NA
Mean values (mmHg)	42 ± 20	
Superior to 30 mmHg, n (%)	16 (29%)	
Left atrial diameter (mm) *	41.3 ± 7.8	35.4 ± 3.2
Mitral regurgitation		
Grade II or above, n (%)	8 (14%)	0 (0%)
Mitral regurgitation related to SAM, n (%)	6 (11%)	0 (0%)
HCM pattern (Maron classification)		NA
Type 1, n (%)	11 (20%)	
Type 2, n (%)	5 (9%)	
Type 3, n (%)	32 (58%)	
Type 4, n (%)	7 (13)	
Apical HCM, n (%)	5 (9%)	NA

HCM, hypertrophic cardiomyopathy; LV, left ventricular; NA, not applicable; SAM, systolic anterior motion. * *p* < 0.001 between HCM group and control group.

**Table 3 jcm-11-02089-t003:** Regional longitudinal strain with standard analysis and strain adjusted for thickness (mid-wall layer).

Longitudinal Strain (%)	Standard	Adjusted	*p* Value
Basal inferoseptal segment	−10.6 ± 6.8	−8.6 ± 6.3	0.0002
Median inferoseptal segment	−14.2 ± 6.5	−12.3 ± 7.1	0.001
Apical inferoseptal segment	−18.1 ± 7.4	−21.4 ± 9.6	0.003
Basal anterolateral segment	−13.8 ± 8.1	−13.8 ± 6.2	0.94
Median anterolateral segment	−13.0 ± 6.9	−13.2 ± 7.4	0.78
Apical anterolateral segment	−15.3 ± 7.2	−18.6 ± 11.3	0.08
Basal anteroseptal segment	−10.7 ± 7.8	−8.4 ± 5.8	0.067
Median anteroseptal segment	−13.7± 6.4	−12.5 ± 6.7	0.02
Apical segment	−16.9 ± 7.1	−18.6 ± 8.4	0.005

## Data Availability

The data presented in this study are available on request from the corresponding author.

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
