# Peer review of "Usefulness of Longitudinal Strain Adjusted to Regional Thickness in Hypertrophic Cardiomyopathy"

_jcm, 2022, doi:10.3390/jcm11082089_

Round 1

Reviewer 1 Report

This manuscript is well written, and the results of the study have potential practical interest. 

I have only one comment:

- The chapter "introduction" seems incomplete. I strongly recommend adding information about the epidemiology (prevalence) and etiology of HCM, classification (types) of cardiomyopathy and pathogenesis. Thus, the chapter "introduction" needs to be significantly expanded.

Author Response

In response to the comments of Reviewer # 1:

“This manuscript is well written, and the results of the study have potential practical interest.”

We thank the reviewer for this positive comment.

  1. “The chapter "introduction" seems incomplete. I strongly recommend adding information about the epidemiology (prevalence) and etiology of HCM, classification (types) of cardiomyopathy and pathogenesis. Thus, the chapter "introduction" needs to be significantly expanded.”

We thank the reviewer. We significantly improved the introduction according to the recommendations of the Reviewer. We added the classification of cardiomyopathies, the prevalence of HCM, the etiology of HCM and clarified the pathogenesis of HCM. In the introduction, we added the following sentences: “Cardiomyopathies are usually divided into four main subtypes [1]: hypertrophic cardiomyopathy, dilated cardiomyopathy, restrictive cardiomyopathy, arrhythmogenic right ventricular cardiomyopathy and other non-classified subtypes.”, “Its prevalence ranges from 0.02% to 0.23% [1].”, “In around 50% of HCM, the disease is an autosomal dominant trait (mutations in cardiac sarcomere protein genes).” and “In 5-10% of cases, other genetic or non-genetic causes may induce HCM, such as glycogen storage diseases, Anderson-Fabry disease, Friedrich’s ataxia, malformation syndromes, amyloidosis, mitochondrial diseases [1]…”. In the following sentence, we added “(hypertension, aortic stenosis)” “Hypertrophic cardiomyopathy (HCM) is a heart disease defined by the presence of increased left ventricular (LV) wall thickness without abnormal loading conditions (hypertension, aortic stenosis) [1-4].”.

Reviewer 2 Report

This is an interesting manuscript focusing on the evaluation of patients with hypertrophic cadiomyopathy by transthoracic ecocardiography and strain.

I have some suggestions for the authors.

  1. Line 110- The authors forgot a French word in the English text: “external border avec the myocardial wall”.
  2. The authors should use the control group in Table 1 and 2 and mention the values of this group in a separate column, marking with “-” or “not applicable” the values which were not calculated for the control group. A p value could be calculated between the values of the two groups, where it is possible.

Author Response

In response to the comments of Reviewer # 2:

“This is an interesting manuscript focusing on the evaluation of patients with hypertrophic cardiomyopathy by transthoracic echocardiography and strain.”

We thank the reviewer for this positive comment.

  1. “Line 110- The authors forgot a French word in the English text: “external border avec the myocardial wall”.”

We apologize for this mistake. We performed the asked modification (delete “avec”, use “with”): “The ROI was manually enlarged to fit internal and external border with the myocardial wall, allowing an analysis of the entire hypertrophic myocardium even if there was asymmetric thickness between the different segments (figure 1B).”

  1. “The authors should use the control group in Table 1 and 2 and mention the values of this group in a separate column, marking with “-” or “not applicable” the values which were not calculated for the control group. A p value could be calculated between the values of the two groups, where it is possible.”

We thank the Reviewer for this important remark. We added the control group in Tables 1 and 2. We modified the title of Table 1: “Table 1. Clinical characteristics of HCM and control groups.”. We also modified the title of Table 2: “Table 2. Echocardiographic characteristics of HCM and control groups.” We added the clinical and echocardiographic characteristics of the control group. Furthermore, we added the significant p values between HCM and control groups in Tables 1 and 2. Significant differences were observed for NYHA classification, left ventricular diastolic diameter, maximal septal thickness, septum/posterior wall ratio and left atrial diameter. All these p values were < 0.001 and were marked by * in the Tables and by *p<0.001 between HCM group and control group below the Tables.
